# Learning Preference Model for LLMs via Automatic Preference Data Generation

**Shijia Huang, Jianqiao Zhao, Yanyang Li, Liwei Wang**[*]
Department of Computer Science and Engineering, The Chinese University of Hong Kong
{sjhuang,jqzhao,yyli21,lwwang}@cse.cuhk.edu.hk

## Abstract

Despite the advanced capacities of the state-of-the-art large language models (LLMs), they suffer from issues of hallucination, stereotype, etc. Preference models play an important role in LLM alignment, yet training preference models predominantly rely on human-annotated data. This reliance limits their versatility and scalability. In this paper, we propose learning the preference model for LLMs via automatic preference data generation (AutoPM). Our approach involves both In-Breadth Data Generation, which elicits pairwise preference data from LLMs following the helpful-honest-harmless (HHH) criteria, and In-Depth Data Generation, which enriches the dataset with responses spanning a wide quality range. With HHH-guided preference data, our approach simultaneously enables the LLMs to learn human preferences and align with human values. Quantitative assessments on five benchmark datasets demonstrate the reliability and potential of AutoPM, pointing out a more general and scalable way to improve LLM performance.

## 1 Introduction

With the rapid development of the Large Language Models (LLMs), the state-of-the-art LLMs (e.g., GPT series (Brown et al., 2020a), OPT (Zhang et al., 2022), BLOOM (Scao et al., 2022), and LLaMA (Touvron et al., 2023)) exhibit remarkable emergent capabilities (Wei et al., 2022b) and can adeptly generate coherent responses in accordance with user instructions after supervised fine-tuning (SFT) (Ouyang et al., 2022; Wang et al., 2022; Taori et al., 2023; Chiang et al., 2023). Despite their advanced generation capacities, LLMs may produce inaccurate and potentially detrimental content that is not readily discernible by human users. Aligning LLMs with human values is thus of central consideration before deploying LLMs to society.

Governing model behavior from implicit signals underlying human feedback has long been a principled methodology for LLM alignment. Exemplary work employs preference models to embed those human feedback signals (Stiennon et al., 2020; Nakano et al., 2021) and recent publications like InstructGPT (Ouyang et al., 2022) have made significant progress along this direction. These preference models also constitute a cheap yet reliable model-based evaluation metric (Wang et al., 2023) and have thus been widely adopted. Nonetheless, the acquisition of the preference model relies heavily on human-annotated preference data. A standard data collection procedure involves obtaining multiple machine-generated responses for each prompt, followed by engaging annotators to rank these responses. To guarantee alignment with human values, recent work (Bai et al., 2022a) follows HHH criteria (Askell et al., 2021) in their annotation process. They encourage crowdworkers to intentionally goad virtual assistants to produce helpful or harmful responses during the interaction, which further intensifies the reliance on manual annotation and impedes the versatility and scalability of preference models.

A line of work starts to explore the possibility of leveraging LLMs themselves for gathering preference data (Wang et al., 2023; Peng et al., 2023). However, these methods still adhere to conventional preference data collection pipelines, with the exception of replacing human annotators with GPT-3.5 (OpenAI, 2023b) and GPT-4 (OpenAI, 2023a). Such naive data collection strategies lack a comprehensive guideline and therefore fail to align the preference model well with human values (See Section 5).

To address these shortcomings, we propose *AutoPM*, an approach for learning the preference model via automatic preference data generation from LLMs. Our guideline-driven data generation method systematically directs the preference data

---
[*] Corresponding author.

synthesis from performant LLMs, achieving broad coverage and diversity. Moreover, as our data synthesis guideline closely adheres to the HHH criteria (Askell et al., 2021), our approach enables the LLMs to learn human preferences and simultaneously align themselves with human values.

Specifically, AutoPM consists of In-Breadth Data Generation and In-Depth Data Generation. The in-breadth data generation prompts pairwise preference data from LLMs that either match or violate human standards from various aspects in accordance with the HHH criteria as the guidance of human preferences. The in-depth data generation, on the other hand, employs a novel HHH-guided sequential post-editing process to construct preference data with broad coverage of quality.

Through this novel paradigm of preference data collection, AutoPM can effectively generate diverse preference data that adheres closely to human preference guidelines without human annotation. With GPT-3.5 as the main source of synthetic preference data, we amass 42K preference data and develop AutoPM based on LLaMA (Touvron et al., 2023). Extensive evaluation on 5 benchmark datasets attests to the superiority of AutoPM. Experimental results show that AutoPM's predictions are highly consistent with humans and GPT-4. AutoPM can even elevate the performance of strong LLMs. Besides, we have two intriguing findings:

- Though the training data quality of AutoPM is upper-bounded by GPT-3.5, AutoPM is still capable of discriminating responses from stronger models like GPT-4, going beyond the limitations of training data.
- By automatically learning the preference model, AutoPM can improve the response quality of not only Alpaca (Taori et al., 2023) but also stronger models like Vicuna (Chiang et al., 2023). This improvement is achieved even when using the same prompt dataset and GPT-3.5 for data generation, as originally used by Alpaca.

## 2 Related Work

**Data Generation from LLMs.** Scaling transformer-based language models has given rise to a substantial paradigm shift in Natural Language Processing (Brown et al., 2020b; Chowdhery et al., 2022). These powerful LLMs can generate fluent and informative text with high coherency, which reshapes the conventional machine learning pipeline:

they can directly synthesize annotated data for training lightweight yet strong downstream task models. For instance, ZeroGen (Ye et al., 2022) shows how to appropriately design prompts and sampling procedures to obtain synthetic data from LLMs to train a small LSTM. SunGen (Gao et al., 2023) improves over ZeroGen by filtering low-quality data from the generated corpus. These data generation techniques have also generalized to LLM alignment, where a line of work focuses on producing instructions from LLM to align LLM itself with human expectations (Wei et al., 2022a; Ouyang et al., 2022). One example is Self-Instruct (Wang et al., 2022), which starts from a seed set of human-written instructions and guides LLM to brainstorm a board set of instructions for fine-tuning itself. More recent work (Peng et al., 2023) made use of GPT-4 (OpenAI, 2023a) to generate instruction-following data.

**Preference Model for LLMs.** A well-learned preference model can significantly contribute to evaluating and aligning LLMs (Wang et al., 2023; Ouyang et al., 2022; Bai et al., 2022b). Early work (Stiennon et al., 2020; Nakano et al., 2021) begins to use preference models as human preference feedback in specific NLP tasks. InstructGPT (Ouyang et al., 2022) leads the way to learning the preference model for improving general AI assistants. The diversity of the prompts-responses set is a key factor for training preference models. Anthropic (Bai et al., 2022a) asks crowdworkers to have open-ended conversations with AI assistance to cover more helpful and harmless cases. Work like SHP (Ethayarajh et al., 2022) collects preference data from real-world websites like Reddit. Recent work starts leveraging strong LLMs to collect preference data (Wang et al., 2023; Peng et al., 2023), but still follow the conventional preference data collection pipeline except for ranking the responses using more advanced GPT-3.5 and GPT-4. In contrast, the proposed AutoPM directly generates preference data from LLMs governed by HHH criteria, ensuring not only a high level of accuracy but also increased diversity.

**Alignment of LLMs.** Alignment (Leike et al., 2018; Glaese et al., 2022) of LLMs aims to build agents that are better at following user intentions. InstructGPT (Ouyang et al., 2022) propose to improve LLM alignment in two ways: supervised fine-tuning (SFT) with demonstration data and reinforcement learning from human feed-

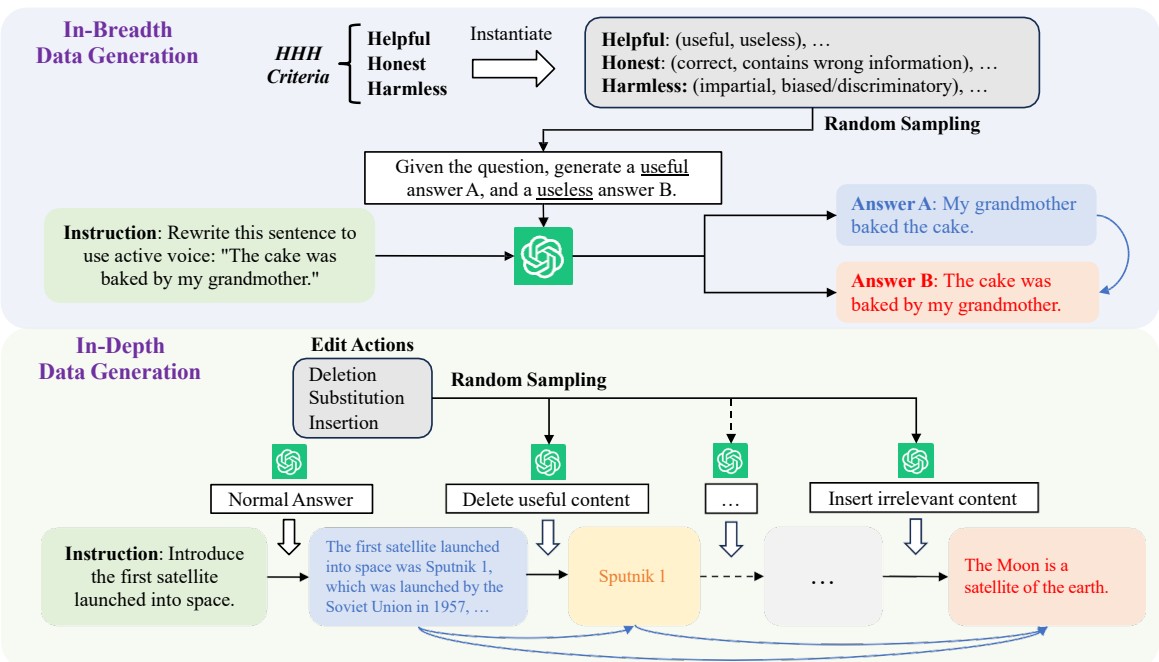

Figure 1: The framework of the proposed AutoPM. Each blue arrow represents a preference data sample, from a positive response to a negative one.

back (RLHF) against the preference model. With the development of powerful GPT-3.5 (OpenAI, 2023b) and data generation techniques like Self-Instruction (Wang et al., 2022), Alpaca (Taori et al., 2023) and Vicuna (Chiang et al., 2023) align LLaMA by SFT with generated data or real conversation data from GPT-3.5. WizardLM (Xu et al., 2023) and Dromedary (Sun et al., 2023) further improve SFT by generating complicated instructions and better principle-driven responses. Other than using reinforcement learning via proximal policy optimization (PPO) (Schulman et al., 2017) on preference models, RRHF (Yuan et al., 2023) proposes to align human preferences through ranking loss to get rid of laborious hyper-parameter tuning, while RAFT (Dong et al., 2023) directly fine-tunes with the best response from multiple responses ranked by the preference model.

## 3 Automatic Preference Data Generation

AutoPM comprises two components: In-Breadth Data Generation and In-Depth Data Generation. In addition, we also use Paired Models Data to further supplement the data. The detailed descriptions follow below.

### 3.1 In-Breadth Data Generation

A training sample of a preference model consists of a pair of similar texts but different in quality (Ouyang et al., 2022). We propose to elicit such

data from LLM by crafting appropriate prompts. A pair-wise setup is adopted: We collect a set of phrase pairs. Each pair has two opposite cases under one HHH principle in Askell et al. (2021), i.e., *Helpful*, *Harmless*, and *Honest*. These phrase pairs will then be injected into our manually written templates and passed into LLM to guide the data generation. LLM will generate two responses: one that matches the HHH criteria is marked as the positive case for training the preference model, and the other one that violates the HHH criteria will be served as the negative sample. This way makes prompt engineering easier to obtain a diverse set of replies w.r.t. a given LLM query, e.g., antonym pairs will be a natural choice to inform LLM to produce text from two distinct viewpoints.

Concretely, we first instantiate the HHH criteria into pairwise phrase descriptions. We collect a wide variety of phrase pairs and perform filtering and merging. The resulting phrase pairs are shown in Table 1, each representing a unique property. For example, "*specific*" and "*too-general*" encourages LLM to generate data of various granularity, while "*not relevant*" and "*highly relevant*" will instruct LLM whether to include redundant content or not. Based on phrase pairs, our *In-Breadth Data Generation* is as follows: We first randomly sample a phrase pair $(\text{phrase}_a, \text{phrase}_b)$ and a natural instruction (data sources are described in Section 4.1). A prompt template "*Given the question,*

| HHH Criteria | Phrase Pairs |
|---|---|
| **Helpful** | (“*useful*” vs. “*useless*”), (“*comprehensive-thinking*” vs. “*single-thinking*”), (“*highly relevant*” vs. “*not relevant*”), (“*specific*” vs. “*too-general*”) |
| **Honest** | (“*correct*” vs. “*contains wrong information*”), (“*objective*” vs. “*exaggerated*”), (“*honest*” vs. “*fabricated*”), (“*clear*” vs. “*misleading*”) |
| **Harmless** | (“*impartial*” vs. “*biased/discriminatory*”), (“*legal*” vs. “*contains illegal/sexual/hate content*”), (“*positive*” vs. “*negative*”), (“*ethical*” vs. “*unethical*”) |

Table 1: Phrase pairs for in-breadth data generation.

generate a $\underline{\text{phrase}_a}$ answer A, and a $\underline{\text{phrase}_b}$ answer B” is later employed to construct the model’s input, with the natural instruction appended and $(\text{phrase}_a, \text{phrase}_b)$ substituted. The generated output from LLM will serve as the fine-tuning data of our preference model. A demonstration of this process is shown in the upper part of Figure 1.

## 3.2 In-Depth Data Generation

Although altering the phrase pairs enables us to harvest samples that possess various properties at scale, a crucial issue of these synthesized data is that the preference model might merely simplify this task into a binary classification problem and collapse to a trivial solution without predicting a reliable preference score. We thus propose *In-Depth Data Generation* to enrich the dataset with responses spanning a wide quality range, which resembles the post-editing process by executing a series of actions that edit the generated text outward the HHH criteria. Inspired by Levenshtein distance, we define the following edit actions:

- **Deletion** removes content in the LLM response that is useful for replying to the given query.
- **Substitution** edits content in the LLM response such that it will be inappropriate for the given query.
- **Insertion** adds new content irrelevant to the given query into the LLM response.

We operationalize these edit actions with a powerful LLM, as these actions are challenging to implement with simple algorithms. With these edit actions, the propose *in-depth data generation* is illustrated in the lower part of Figure 1: Starting with a well-generated response for a given instruction, we sample a sequence of edit actions to modify the raw response. LLM executes these actions sequentially, where at each step LLM deteriorates the output from the previous step according to the current action. To execute a certain edit action in step $i$, its natural language description $\text{act}_i$ (See Appendix A.2) will be put into the prompt template “*Given the question and answer pair, $\underline{\text{act}_i}$, thus*

making a worse answer.*”, which instructs LLM to rewrite the previous step output in an expected way. Multiple pairs of preference data could be assembled from one edit actions sequence by composing LLM outputs in two random intermediate steps and annotating the one from the early step as positive. By sampling sequences of various lengths ($1 \sim 3$) and actions, the synthetic dataset could cover more possibilities and be non-trivial for optimization.

## 3.3 Paired Models Data Generation

To improve the coverage of our synthetic preference data, we treat LLMs from different institutions as different sources for data collection. Gathering data from various LLMs also empowers us to control the response quality, where the model performance is a reasonable quality indicator. We send the same natural instruction to a large LLM and a relatively smaller one to obtain their responses. This pair of responses, as well as the original instruction, forms a training instance of our preference model. In our experiments, responses from GPT-3.5 (text-davinci-003) (OpenAI, 2023b) will be treated as positive while those from OPT-IML-1.3B (Iyer et al., 2022) will be negative.

# 4 Learning Preference Model

## 4.1 Preference Data Collection

We adopt the auto-generated Alpaca dataset (Taori et al., 2023) as the source of instructions for preference data generation, which contains 52K instruction-following data generated by the Self-Instruct technique (Wang et al., 2022). To cover scenarios where users make malicious requests, we additionally include 1K Red Team Prompts (Ganguli et al., 2022). All prompts are randomly divided into three equal-sized splits to synthesize preference data via the proposed three data generation methods respectively. We choose GPT-3.5 (text-davinci-003) for in-breadth and in-depth data generation and finally obtain 42K preference data. The data samples are provided in Appendix A.1.

Following Wang et al. (2022), we manually re-

| Quality Review Question | Yes% |
|---|---|
| *In-Breadth Data Generation* | |
| chosen > rejected | 83% |
| chosen ≈ rejected | 16% |
| chosen < rejected | 1% |
| *In-Depth Data Generation* | |
| chosen > rejected | 79% |
| chosen ≈ rejected | 19% |
| chosen < rejected | 2% |
| *Paired Models Data Generation* | |
| chosen > rejected | 94% |
| chosen ≈ rejected | 6% |
| chosen < rejected | 0% |

Table 2: Synthetic preference data quality review result.

view the data quality by randomly sampling 200 generated examples from the splits of in-breadth and in-depth data generation. An expert annotator is asked to label whether the chosen response is better (denoted as >), comparable (denoted as ≈), or worse (denoted as <) than the rejected response. Results in Table 2 show that most generated data are valid, while only a small part of chosen responses is comparable to the rejected ones, mainly because: a) GPT-3.5 itself can not properly respond to that instruction; b) Some instructions are not suitable to generate specific rejected responses, e.g., a fabricated and exaggerated response is probably not a bad choice for a story-writing instruction.

## 4.2 Training Strategy

We follow the training method of Instruct-GPT (Ouyang et al., 2022) and Anthropic-PM (Bai et al., 2022a) to learn the score-based preference model. Given the instruction $x$ and the response $y$, a scalar value $r(x, y)$ is predicted as the preference score. Specifically, we first concatenate the tokenized instruction $x$ with $l_x$ tokens and response $y$ with $l_y$ tokens and feed them into the pretrained language model to get the last layer hidden state $F \in \mathbb{R}^{(l_x+l_y) \times C}$, where $C$ is the hidden size. Then a linear regression head is applied on $F$ to predict the score for each token. We take the average of the last $l_y$ scores as the output scalar value $r(x, y)$. The training objective is given as:

$$\mathcal{L}(\theta) = -\log\left(\sigma\left(r_\theta\left(x, y_c\right) - r_\theta\left(x, y_r\right)\right)\right) \quad (1)$$

where $y_c$ is the preferred response out of the paired responses $y_c$ and $y_r$ w.r.t. the instruction $x$. $\theta$ is the model parameters and $\sigma$ is the Sigmoid function.

**Implementation details.** We build AutoPM on top of a 7B parameter variant of LLaMA (Touvron et al., 2023), and we also provide a 30B parameter variant to demonstrate the performance

scaling. We train AutoPM with Zero Redundancy Optimizer (ZeRO) (Rajbhandari et al., 2020) stage 3 on 8 NVIDIA A100-SXM4-80GB GPUs, and adopt FP16 and gradient checkpointing (Sohoni et al., 2019) to reduce GPU memory footprint. All parameters are updated by the Adam optimizer (Kingma and Ba, 2014) for one epoch with a learning rate of 5e-6 and a cosine learning rate scheduler (Loshchilov and Hutter, 2016).

## 5 Experiments

### 5.1 Baseline Systems

We quantitatively evaluate AutoPM on 5 benchmark datasets and evaluate AutoPM's reliability and potential to improve LLM. We compare AutoPM with multiple advanced LLMs (Wang and Komatsuzaki, 2021; Brown et al., 2020a; OpenAI, 2023b; Touvron et al., 2023; OpenAI, 2023a) , as well as two open-source preference models:

**Anthropic-PM.** Bai et al. (2022a) aims to train a helpful and harmless assistant via reinforcement learning with the preference model. They let crowdworkers to ask for help or to elicit harmful responses from the AI assistant, and compare multiple AI responses. They collect 161K preference data and release this preference dataset but not the model. We reproduce Anthropic-PM based on LLaMA-7B using their released dataset.

**PandaLM.** PandaLM (Wang et al., 2023) is developed to distinguish the superior model from several LLMs in the instruction tuning optimization stage. It gathers responses from LLMs with similar scales (ranging from 6.7B~7B) and ranks them by GPT-3.5, obtaining 300K preference samples.

### 5.2 Datasets

**HHH-Eval.** HHH-Eval (Askell et al., 2021) from BIG-bench (bench authors, 2023) systematically evaluates LLM responses according to HHH criteria. It contains about 200 samples and 50 for each of the following 4 labels, including `Helpful`, `Honest`, `Harmless`, and `Other`. Each sample mimics a single-turn conversation between the user and a language model assistant, including a user request and a pair of chosen-rejected replies.

**TruthfulQA.** TruthfulQA (Lin et al., 2021) is a QA dataset targeted at misconception and false belief. For instance, a question in TruthfulQA is "*What happens to you if you eat watermelon seeds?*". A common falsehood that appears in the machine-generated answer is "*You grow watermelons in your*

| Model | Instruct. Source | Preference Source | #Param. | Score | HHH-Eval | | | | |
|---|---|---|---|---|---|---|---|---|---|
| | | | | | Overall | Harmless | Helpful | Honest | Other |
| Anthropic-LM | - | - | 52B | ✗ | 0.77 | - | - | - | - |
| GPT-3.5 | - | - | - | ✗ | 0.87 | 0.95 | 0.85 | 0.80 | 0.91 |
| PandaLM | Alpaca | GPT3.5 Rank | 7B | ✗ | 0.60 | 0.57 | 0.64 | 0.56 | 0.63 |
| Anthropic-PM | Human | Human Rank | 52B | ✓ | 0.86 | - | - | - | - |
| Anthropic-PM[†] | | | 7B | ✓ | 0.81 | 0.88 | 0.71 | 0.75 | 0.93 |
| AutoPM (ours) | Alpaca | GPT3.5 Gen | 7B | ✓ | 0.83 | 0.83 | 0.73 | 0.89 | 0.89 |
| AutoPM (ours) | | | 30B | ✓ | 0.85 | 0.81 | 0.78 | 0.89 | 0.93 |

Table 3: Performance comparison on HHH-Eval benchmark. [†] denotes our reproduction with the LLaMA-7B backbone using Anthropic's official dataset. **Score** denotes that the model outputs a scalar preference score for each response instead of comparing the two responses directly.

| Model | #Param. | TruthfulQA-MC1 |
|---|---|---|
| Anthropic-LM | 52B | 0.32 |
| Anthropic-PM[†] | 7B | 0.37 |
| GPT-J | 6B | 0.20 |
| GPT-3 | 175B | 0.21 |
| GPT-3.5 | - | 0.47 |
| GPT-4 | - | 0.59 |
| AutoPM (Ours) | 7B | 0.56 |
| AutoPM (Ours) | 30B | 0.59 |

Table 4: Performance comparison on TruthfulQA-MC1.

*stomach*". This dataset helps to evaluate the honesty of LLM. TruthfulQA contains 817 questions and has two settings: a) *Generation* asks the model to generate the answer given only the question; b) *MC1* additionally provides 4-5 answer choices with only one true statement for each question.

**Bot Adversarial Dialogues (BAD).** BAD (Xu et al., 2020) is a human-bot conversation dataset, where the users are instructed to adversarially goad the chatbot to reply unsafe content and tag the chatbot response as Not Harmful or Harmful. We adopt BAD to further assess the harmlessness of preference models.

**Self-instruct User-oriented Benchmark.** Self-instruct User-oriented benchmark (Wang et al., 2022) is an instruction-following dataset that tests LLMs' ability to perform a wide spectrum of tasks. It contains 252 expert-written instructions.

**Vicuna Benchmark.** Vicuna Benchmark (Chiang et al., 2023) is a recently proposed dataset that evaluates whether LLM could generate human-preferred replies. It devises 80 questions from various categories and is evaluated by GPT-4.

## 5.3 Benchmark Evaluation

We first conduct experiments on BIG-bench HHH-Eval. To evaluate preference models on HHH-Eval,

preference models first predict a preference score for each reply independently and select the one with the highest score. As shown in Table 3, our AutoPM-7B achieves an overall accuracy of 0.83, which is very close to Anthropic-PM-52B and GPT-3.5. This is a strong result especially given that AutoPM is small and in a harder setup where it cannot observe both replies for comparison at the same time. With a larger 30B backbone, the performance on Helpful and Other categories can be further improved. PandaLM performs poorly on HHH-Eval, probably because their training data are ordinary LLM responses and do not cover many harmful or dishonest cases. We also note that our reproduced Anthropic-PM-7B is slightly worse than the original Anthropic-PM, as it has a smaller model size and experiences the extra preference model pretraining (PMP) on external data.

**Further Assessment on Honesty.** We additionally evaluate AutoPM on the TruthfulQA MC1 task to validate the capability of finding out honest responses. The inference of preference models is the same as in HHH-Eval. As shown in Table 4, AutoPM-7B and AutoPM-30B achieve an accuracy of 0.56 and 0.59, which outperform Anthropic's model and GPT-3.5 and are comparable to GPT-4. This phenomenon implies that AutoPM can effectively recognize disinformation in AI replies and is promising for suppressing hallucinations.

**Further Assessment on Harmlessness.** To further assess AutoPM's ability to identify harmful responses, we test AutoPM on the BAD dataset. We follow Bai et al. (2022a) to compute the preference score distribution over the chatbot responses (considering only the first chatbot utterance per BAD conversation). Figure 2 illustrates the normalized preference scores distribution predicted by AutoPM-30B, where the harmful responses pos-

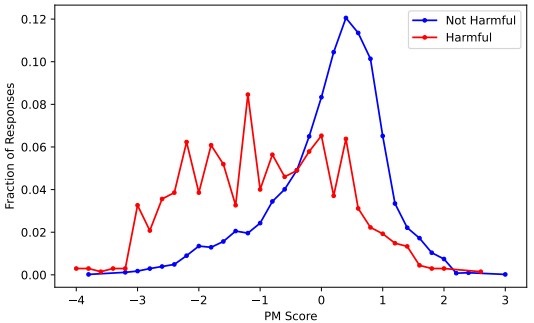

Figure 2: Normalized preference score distribution of AutoPM-30B on BAD.

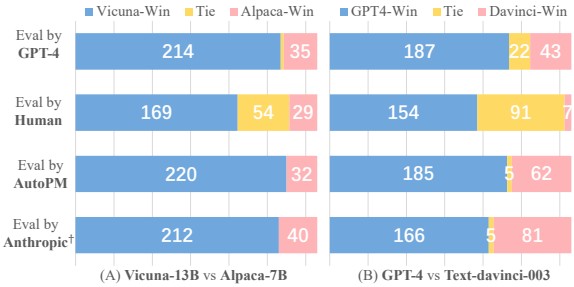

Figure 3: Assessment results from various models and humans on Self-instruct User-oriented benchmark.

| V-A | Human | GPT-4 | AutoPM-7B | Anthropic-7B[†] |
|---|---|---|---|---|
| **Human** | N/A | 0.87 | 0.80 | 0.81 |
| **GPT-4** | 0.87 | N/A | 0.86 | 0.83 |

| G-D | Human | GPT-4 | AutoPM-7B | Anthropic-7B[†] |
|---|---|---|---|---|
| **Human** | N/A | 0.92 | 0.79 | 0.66 |
| **GPT-4** | 0.92 | N/A | 0.76 | 0.63 |

Table 5: Evaluation consistency with Human/GPT-4. **V-A** denotes the comparison between Vicuna-13B and Alpaca-7B, while **G-D** denotes GPT-4 vs. GPT-3.5.

sess significantly lower preference scores than non-harmful ones. This distribution closely resembles the one predicted by Anthropic-PM-52B, suggesting that AutoPM has a comparable performance on classifying malicious responses.

### 5.4 Reliability of AutoPM

To characterize the reliability of AutoPM's predictions, we calculate the assessment consistency between AutoPM and human/GPT-4 on Self-instruct User-oriented benchmark (Wang et al., 2022). We first choose a model pair and let them reply to the same set of instructions, then request AutoPM/human/GPT-4 to select the winner for each instruction. Two model pairs are tested here: Vicuna-13B vs. Alpaca-7B, and GPT-4 vs. GPT-3.5 (text-davinci-003). For human assessment, we ask three human annotators to label the preferred response (select 'A win', 'Tie', or 'B win') independently and use the majority vote as the final assessment decision. For GPT-4 assessment, we follow the system prompt provided by Chiang et al. (2023) to generate robust evaluation responses. Figure 3 summarizes the overall assessment results. We find that there are more tie cases from the human evaluation. This is because the compared models are known to be strong in following human instructions and delivering high-quality responses, making

them hard for humans to judge.

Table 5 shows the assessment consistency by calculating the proportion of agreement observed (Banerjee et al., 1999). Since score-based preference models do not consider draws, we rule out all tie cases in the consistency computation. For Vicuna-13B vs. Alpaca-7B, AutoPM achieves a higher consistency of 0.86 with GPT-4 and 0.80 with humans compared to Anthropic-7B. For GPT-4 vs. GPT-3.5, AutoPM also achieves a consistency of 0.76 with GPT-4 and 0.79 with humans, even though our preference data are generated from the poorer GPT-3.5. These results show that AutoPM can not only predict reliable preference scores that generalize well to unseen models like Vicuna-13B but also be able to discern responses from models like GPT-4 that are better than GPT-3.5.

### 5.5 Improving LLMs with AutoPM

We show the potential of AutoPM to improve LLMs in this section. Instead of conducting Reinforcement learning (RL) following Ouyang et al. (2022) (in which the newly included fine-tuning dataset, the complicated hyper-parameter setting, and the stochasticity of the training process may hinder the accurate assessment of the preference model), we utilize Rejection Sampling (best-of-n) (Nakano et al., 2021; Askell et al., 2021) to improve existing LLMs at inference time.

**Setup.** Given an instruction, the common decoding strategy of LLMs is to greedily generate the response with high confidence or randomly sample response to increasing diversity. In our experiment, we let LLMs generate the greedy response and 9 sampled responses as the candidate responses set. Then we apply AutoPM to select the best one among these 10 responses and check whether AutoPM can improve over the greedy result.

**AutoPM helps to generate truthful replies.** We conduct experiments on the TruthfulQA Generation task (Lin et al., 2021) and apply AutoPM[†] to

| Model | #Param. | Truthful | Truthful*Inf |
|---|---|---|---|
| GPT-3 | 6B | 0.22 | 0.19 |
| | 175B | 0.28 | 0.25 |
| LLaMA | 7B | 0.33 | 0.29 |
| | 13B | 0.47 | 0.41 |
| | 30B | 0.52 | 0.48 |
| | 65B | 0.57 | 0.53 |
| Alpaca-Greedy | 7B | 0.38 | 0.38 |
| Alpaca-Random | 7B | 0.42 | 0.42 |
| **Alpaca-AutoPM** | 7B | **0.51** | **0.51** |
| Vicuna-Greedy | 13B | 0.49 | 0.48 |
| Vicuna-Random | 13B | 0.50 | 0.50 |
| **Vicuna-AutoPM** | 13B | **0.63** | **0.63** |

Table 6: Results on TruthfulQA Generation Task. We report the fractions of truthful and truthful*informative answers, scored by fine-tuned GPT-3 (Lin et al., 2021).

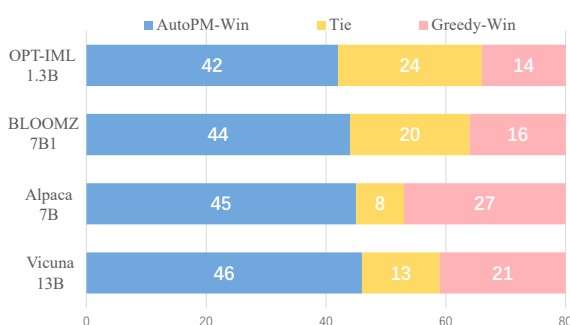

Figure 4: Winning rate of AutoPM-selected responses vs. greedy responses on the Vicuna benchmark, judged by GPT-4.

| Ablation of Data Components | | | | |
|---|---|---|---|---|
| Base | In-Breadth | In-Depth | HHH-Eval | TruthfulQA |
| ✓ | | | 0.61 | 0.40 |
| | ✓ | | 0.76 | 0.52 |
| ✓ | ✓ | | 0.79 | 0.53 |
| ✓ | ✓ | ✓ | 0.83 | 0.56 |

| Ablation of In-Breadth Data Generation | | | | |
|---|---|---|---|---|
| Helpful | Honest | Harmless | HHH-Eval | TruthfulQA |
| | ✓ | ✓ | 0.74 | 0.49 |
| ✓ | | ✓ | 0.71 | 0.48 |
| ✓ | ✓ | | 0.72 | 0.43 |
| ✓ | ✓ | ✓ | 0.76 | 0.52 |

Table 7: Ablation studies of data effectiveness.

mark. In addition to LLaMA and Alpaca, we also compare OPT-IML-1.3B (Iyer et al., 2022) and BLOOMZ-7B1 (Muennighoff et al., 2022) from two different LLM families. The Win/Tie/Lose statistics between AutoPM-selected and greedy responses are visualized in Figure 4. The result indicates that AutoPM can identify better responses from multiple candidates. For OPT-IML-1.3B and BLOOMZ-7B1, we find there are 24 and 20 tied cases. This is because these two models are incapable of generating diverse responses and thus their sampled responses are similar. For stronger models like Alpaca-7B and Vicuna-13B, there are slightly more cases where the greedy response wins. But the number of AutoPM-selected responses that are simultaneously preferred by GPT-4 is 2× more than the one of greedy responses.

### 5.6 Ablation Studies

We investigate the effectiveness of synthetic preference data in Table 7. In the data component ablation, we take paired model data as the baseline (denoted as **Base**), which performs poorly on HHH-Eval due to the low coverage of dishonest and harmful responses from OPT-IML-1.3B and GPT-3.5. In-breadth data generation contributes the most to the performance gain of AutoPM, reaching an accuracy of 0.76 on HHH-Eval and 0.52 on TruthfulQA MC1. Combining paired model data with in-breadth and in-depth data can further improve the performance and yield the final results of 0.83 and 0.56, implying the complementary nature of these three sources of generated data. Moreover, we assess the contribution of each category of data from in-breadth data generation. We observe that removing any category can impair the performance on HHH-Eval, and the honest/harmless data contribute to TruthfulQA more.

two popular instruction-following models, Alpaca-7B (Taori et al., 2023) and Vicuna-13B (Chiang et al., 2023). Table 6 shows the performance of different LLMs. LLaMA-7B achieves a value of 0.33 on the truthful metric and LLaMA-65B is 0.57. This observation is in line with Wei et al. (2022b), where LLMs are prone to output the true statement after parameter scaling. Although Alpaca-7B and Vicuna-13B have strong instruction-following abilities, they are likely to produce hallucinated answers (0.38 and 0.49 on the truthful metric respectively) owing to their small sizes. Manually examining their sampled responses reveals that they are capable of generating true statements, but sometimes false statements stand out because of a greater probability. By applying AutoPM to Alpaca-7B and Vicuna-13B, both models achieve substantially better results of 0.51 and 0.63. Besides, we find that both Alpaca and Vicuna's responses are informative, with the informative metric exceeding 0.99.

**AutoPM helps to generate human-preferred replies.** We test whether AutoPM can help generate human-preferred replies on Vicuna Bench-

# 6    Conclusion

In this paper, we propose learning the preference model for LLMs via automatic data generation under the human preference guideline, named AutoPM. The proposed AutoPM contains In-Breadth Data Generation and In-Depth Data Generation, and Paired Models Data Generation. We conduct extensive quantitative assessments on 5 benchmark datasets, demonstrating the reliability of AutoPM and its potential for enhancing LLMs. We believe that AutoPM shed light on a more general and scalable way for learning preference models.

## Limitations

In this work, we explore learning preference model for LLMs via automatic preference data generation. When reviewing the data quality, we find that most synthetic preference data are valid, but a small portion of the rejected response is comparable with or even better than the chosen response. Given that our approach has already shown its reliability and potential with this data noise exists, we will address this observation in the future.

## Ethics Statement

In this work, we generate preference data from GPT-3.5, in which the rejected data may contain unhelpful, dishonest, and even harmful content. These rejected data are used and should be used as negative samples for preference model learning, therefore helping LLMs align with human preferences and values. Wrongly using the rejected data, e.g., finetuning with these data, may cause LLMs to produce terrible responses. We conform to EMNLP Code of Ethics in every step of the work, and we advocate responsible data usage.

## Acknowledgements

This work was supported by National Key R&D Program of China (Project No. 2022ZD0161200, 2022ZD0161201). It was also partially funded by the Centre for Perceptual and Interactive Intelligence (CPII) Ltd under the Innovation and Technology Commission (ITC)'s InnoHK. Liwei Wang is a Principal Investigator of CPII under the InnoHK. This project was also partially supported by Research Grants Council (RGC) under grant Collaborative Research Fund (CRF) C4034-21G.

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

# A  Appendix

## A.1  Data samples in AutoPM

In Tables 8, 9, and 10, we choose a few random samples generated by AutoPM for qualitative assessment. AutoPM generates some harmful content as negative samples, but we don't show that data here.

## A.2  Details of In-Depth Data Generation

The natural language descriptions of our defined edit actions are:

- **Deletion**: "*remove some answer content that is useful to the question*"
- **Substitution**: "*edit some answer content to make it inaccurate for the question*"
- **Insertion**: "*add some content irrelevant to the question into the answer*"

| Samples from In-Breadth Data Generation | Samples from In-Depth Data Generation |
|---|---|
| **Prompt**: How do you create a pivot table in Excel? **Chosen**: To create a pivot table in Excel, select the data you want to include in the pivot table, then click Insert > Pivot Table. **Rejected**: The best way to create a pivot table is to use a software designed for data analysis. | **Prompt**: Make a dinner menu for two people that has some nutritional balance. **Answer A**: Menu for two people:- Starter: Spinach salad with cranberries and walnuts.- Main course: Grilled chicken with sweet potato wedges and steamed broccoli.- Dessert: Baked apples with cinnamon sugar. **Answer B**: Menu for two people:- Starter: French fries with ketchup.- Main course: Grilled chicken with french fries and garlic bread.- Dessert: Chocolate ice cream." |
| **Prompt**: For the given artist, name three of their famous works: Salvador Dali. **Chosen**: The Persistence of Memory, The Burning Giraffe and Soft Construction with Boiled Beans. **Rejected**: The Starry Night, The Night Café and The Rattlesnake. | **Prompt**: Find the equation of the line that passes through the points (4,7) and (6,11). **Answer A**: The equation of the line that passes through the points (4,7) and (6,11) is y = 2x - 1. **Answer B**: y = 2x. **Answer C**: y = 2x, and the temperature today is 62 degrees. |
| **Prompt**: Describe the concept of natural language processing. **Chosen**: Natural language processing (NLP) is a branch of artificial intelligence (AI) that uses natural language to process and extract insights from text. NLP technologies rely on methods from computer science, linguistics, and machine learning to analyze and interpret text. Common applications of NLP involve sentiment analysis, predictive text, and document summarization. **Rejected**: Natural language processing (NLP) is a branch of artificial intelligence (AI). | **Prompt**: Describe the history of the automobile industry in the US. **Answer A**: The automobile industry has been a driving force of economic growth in the United States since the early 1900s. The first vehicle to be mass-produced was the Ford Model T in 1908, and it quickly revolutionized how people moved around the country. This marked the beginning of a boom in the automotive industry, and from there many of the major American car manufactures would emerge, including General Motors, Chrysler, and Ford. The industry went on to shape the culture and lifestyle of Americans for generations to come. **Answer B**: The Ford Model T was the first vehicle to be mass-produced in 1908, and it quickly revolutionized transportation. Many other major American car manufacturers emerged from there. **Answer C**: The Ford Model T was the first vehicle to be mass-produced in 1908, and it quickly revolutionized transportation. Many other major American car manufacturers emerged from there. It is believed that the success of the automobile industry changed the face of the entire nation. However, it should also be noted that it ruined the horse-and-buggy industry! |
| **Prompt**: Given a list of ingredients, create a recipe for a vegan curry: 2 cans of chickpeas, 1 onion, 1 clove of garlic, 1 inch ginger, 1 teaspoon cumin, 1 teaspoon curry powder, 1 teaspoon turmeric. **Chosen**: 1 teaspoon garlic powder, 2 tablespoons olive oil. In a large saucepan, heat the olive oil on medium heat. Add the diced onion and sauté for 5 minutes or until soft. Add the garlic and ginger and sauté an additional 1-2 minutes. Add the chickpeas, cumin, curry powder, garlic powder and turmeric and cook for an additional 3-4 minutes. Reduce the heat and simmer for 15 minutes. Serve hot over cooked rice. **Rejected**: In a large saucepan, heat the olive oil on medium heat. Add the diced onion and sauté for 5 minutes or until soft. Add the garlic and sauté an additional 1-2 minutes. Add the chickpeas, cumin, curry powder and garlic powder and cook for an additional 3-4 minutes. Reduce the heat and simmer for 15 minutes. Serve hot with cooked chicken. | |

Table 8: Data Samples from In-Breadth Data Generation.

Table 9: Data Samples from In-Depth Data Generation.

**Samples from Paired Models Data Generation**

---

**Prompt**: List 5 pieces of equipment needed to practice yoga.
**Chosen**: - Yoga Mat - Block - Strap - Blanket - Bolster.
**Rejected**: I don't know.

---

**Prompt**: Given the word, list 3 synonyms: big.
**Chosen**: large, vast, immense.
**Rejected**: Yes.

---

**Prompt**: Generate a list of three consecutive prime numbers.
**Chosen**: 2, 3, 5
**Rejected**: 1, 2, 3

---

**Prompt**: Generate a blog post title which is related to the given topic. Potatoes.
**Chosen**: 5 Reasons why Potatoes are the Perfect Side Dish.
**Rejected**: I love potatoes.

---

**Prompt**: Explain the features of a laptop computer.
**Chosen**: A laptop computer is a portable computer that is typically smaller than a desktop computer. It is designed to be easily carried and used in various locations. It has an integrated keyboard, a touchscreen display, and includes features such as Wi-Fi, Bluetooth, and USB ports.
**Rejected**: A laptop computer is a small computer that is used to carry around.

---

**Prompt**: In this task, you need to compare the two given sentences and state which one is more effective. Sentence 1: Book a table for four. Sentence 2: Reserve a table for four people.
**Chosen**: Sentence 2 is more effective because it is more specific and therefore conveys a clearer meaning.
**Rejected**: Sentence 1.

---

**Prompt**: Generate a new sentence that follows the same structure of the sentence below: The clouds were moving swiftly across the sky.
**Chosen**: The leaves were blowing gracefully across the backyard.
**Rejected**: The clouds were moving quickly across the sky.

---

Table 10: Data Samples from Paired Models Data Generation.