# OpenReview forum: "Learning Preference Model for LLMs via Automatic Preference Data Generation"
_EMNLP/2023/Conference — EMNLP 2023 Main_

### Official Review · Reviewer_Dan7 · 2023-07-31

**Soundness:** 4

**Excitement:**

3: Ambivalent: It has merits (e.g., it reports state-of-the-art results, the idea is nice), but there are key weaknesses (e.g., it describes incremental work), and it can significantly benefit from another round of revision. However, I won't object to accepting it if my co-reviewers champion it.

**Paper Topic And Main Contributions:**

The paper presents  proposes an innovative approach, AutoPM, aimed at enhancing preference model generation by employing two novel data generation methods: In-Breadth Data Generation and In-Depth Data Generation.  These methods collectively contribute to the development of a more reliable preference model, effectively capturing human preferences. The authors back their claims with comprehensive experiments conducted on five benchmark datasets, which further solidify the credibility of their study.

**Reasons To Accept:**

1. The paper's primary contribution, centered around the data generation methods and their impact on Preference Model generation, is clearly defined. The authors meticulously explain the rationale behind each data generation technique, highlighting how it addresses the core problem of capturing human preferences more effectively.
2. The extensive experimentation on five diverse benchmark datasets enhances the credibility of the proposed approach. The authors provide detailed insights into the experimental setup, including the selection of datasets, evaluation metrics, and comparative analyses against existing methods.

**Reasons To Reject:**

While the paper excels in introducing novel data generation methods, some aspects, particularly concerning the training of Preference Models (PM), warrant further exploration. Readers would benefit from a deeper investigation into potential improvements or alternative approaches in training PM models, which could complement the presented data generation techniques. Integrating such discussions would enrich the overall contribution of the paper.

**Reproducibility:**

4: Could mostly reproduce the results, but there may be some variation because of sample variance or minor variations in their interpretation of the protocol or method.

**Reviewer Confidence:**

3: Pretty sure, but there's a chance I missed something. Although I have a good feel for this area in general, I did not carefully check the paper's details, e.g., the math, experimental design, or novelty.

---

> ### Author Rebuttal · Authors · 2023-08-27
>
> We are grateful to reviewer **Dan7** for recognizing our work, as well as the valuable advice. We present our responses in the following.
>
> ### **Q1: Integrating a deeper investigation into potential improvements or alternative approaches in training preference models would enrich the overall contribution of the paper.**
>
> We appreciate your insightful comment. We concur that training techniques play a crucial role in obtaining a performant preference model. However, the main focus of this work is on how to effectively and efficiently collect preference data for training a high-quality preference model. We adopt the same training settings as previous work, such as InstructGPT/AnthropicPM. The exploration of more effective preference model training is out of the scope of this paper and left for future work. We believe that our data synthesis approach can benefit from more powerful training methods, as it demonstrates promising improvement in a general training setup.

---

### Official Review · Reviewer_D9n7 · 2023-08-03

**Soundness:** 4

**Excitement:**

4: Strong: This paper deepens the understanding of some phenomenon or lowers the barriers to an existing research direction.

**Paper Topic And Main Contributions:**

This paper proposes a mechanism for automatically constructing a preference model dataset using a seed dataset of prompts (in practice, the Alpaca dataset prompts) and access to a set of LLMs that can generate answers given those prompts. They consider three strategies for generating preference data from LLMs:

- In-breadth data generation (IBDG): They construct a set of word-antonym pairs associated with each of the HHH criteria (e.g., for helpful an we might have “useful” and “useless”), and prompt an LLM to respond to an instruction with an answer that is characterized by each word in the pair. This creates a set of paired responses to an instruction, one being preferred over the other as it better respects the HHH criteria (at least along one axis).
- In-depth data generation (IDDG): They take an (instruction, response) pair generated from IBDG and prompt an LLM to iteratively degrade it. This creates a sequence of responses, each less preferred than the last.
- Paired models data generation: Given a more powerful LLM and a less powerful LLM, they ask each to generate a response to an instruction, and always assume the more powerful LLMs’ response is preferred.

They use this data to train a preference model (AutoPM), and compare it to openly available preference models as well as off-the-shelf LLMs, when prompted to act as preference models. Their preference model attains competitive performance in many cases. They also show that their preference model can be used to re-rank LLM generations to obtain better answers than greedy decoding (in a rejection sampling setup).

Overall, I think this is a good paper, though it would benefit from improved writing and a few more experiments.

**Questions For The Authors:**

Question A: For IDDG, at the end of the section you say that “Multiple pairs of preference data *could* be assembled […]”, but it is not clear whether you actually do this assembly (and if so, how). Could you clarify this?

Question B: Are the examples in Appendix A.1 sampled uniformly at random? If not, how did you select them?

Question C: Which word-antonym pairs and insert/substitute/delete operations were applied for each of the generations in Appendix A.1?

Question D: In Figure 3 how can you get ties between 3 humans if they only pick winners (as implied by Section 5.4)? Can they also pick ties? Or do they assign scores to each datapoint?

**Reasons To Accept:**

- The authors propose multiple techniques that, when combined, can be used to effectively create a preference dataset with no human annotation (the only manual effort that is required is constructing/collecting the seed prompts)
- The results are strong, with their model trained on preference data being generally competitive and occasionally surpassing openly available preference models (some much larger than theirs) and LLMs when prompted to behave as preference models.
- The experiments are generally solid and do a good job at supporting the validity of their approach. That said, there are some experiments and numbers that would further strengthen the paper (see reasons to reject).

**Reasons To Reject:**

- The writing is frequently unclear. I could still follow the text, but it was often hard to understand exactly what the authors meant. I would encourage the authors to spend some time proof-reading the paper.
- There are some experiments that would further strengthen the paper, for example:
    - It would be good to report inter-annotator agreement for the consistency evaluation in Table 5
    - Ultimately, preference models are most useful for RLHF, so it would be useful to explore how AutoPM compares to your AnthropicPM replication when used for RLHF, specifically
    - For the rejection sampling experiments, it would be good to report numbers using your AnthropicPM replication in addition to the AutoPM numbers.
    - It is not clear to me why a human assessment of the paired model data generation technique is not reported in Table 2.

**Reproducibility:**

4: Could mostly reproduce the results, but there may be some variation because of sample variance or minor variations in their interpretation of the protocol or method.

**Reviewer Confidence:**

3: Pretty sure, but there's a chance I missed something. Although I have a good feel for this area in general, I did not carefully check the paper's details, e.g., the math, experimental design, or novelty.

**Typos Grammar Style And Presentation Improvements:**

In your phrase pairs in Table 1, “contains [wrongly] illegal/…” sounds off. Dropping that word would probably sound better.

There are many points in the paper that would benefit from improved writing. Here are some of them:

L026: “Large Language Model[s]”

L127: “paradigm shift” (not shifting)

L238: “datasources are [described] in” (not depicted)

L245: “a demonstration” (not live demonstration)

L333: “can not [properly respond to that instruction]” (well-respond sounds weird)

L447: “probably” not “properly”

L451: “as it has a smaller size and does not experience the extra …” (since you are referring to the Anthropic-PM-7B model)

---

> ### Author Rebuttal · Authors · 2023-08-27
>
> We are grateful to reviewer **D9n7** for recognizing our work, as well as the valuable advice. We present our responses in the following.
>
>
> ### **R1: Encourage the authors to proofread the paper.**
>
> Thanks for your careful reading and advice to improve the paper's quality, we carefully reread our article and revised some statements to make it more straightforward. We also fixed the typos and grammatical errors you mentioned in *Typos Grammar Style And Presentation Improvements*.
>
> ---
>
> ### **R2: Advice to report inter-annotator agreement for the consistency evaluation in Table 5.**
>
> | Agreement | Annotator 1 | Annotator 2  | Annotator 3 |
> | :----: | :----: | :----: | :----: |
> | **Annotator 1** |  1.0  | 0.89  | 0.90 |
> | **Annotator 2** | 0.89  | 1.0 | 0.81 |
> | **Annotator 3** | 0.90 | 0.81 | 1.0 |
>
> Thanks for your advice. Here we provide the inter-annotator agreement assessment results. We follow PandaLM[1] to calculate the pairwise annotator agreement, considering the binary win-lose cases. Since the human-annotated tag distribution is imbalanced (Vicuna/GPT-4 is much stronger than Alpaca/GPT3.5), we directly report the *relative observed agreement among raters* in Cohen's Kappa.
>
> As shown in Table 1, the assessment consistency of different annotators is high, indicating that our human assessment is reliable.
>
> [1] Wang, Yidong, et al. "PandaLM: An Automatic Evaluation Benchmark for LLM Instruction Tuning Optimization." arXiv preprint arXiv:2306.05087 (2023).
>
> ---
>
> ### **R3: Advice for more experimental comparisons between AutoPM and AnthropicPM.**
>
> We appreciate your constructive feedback. As stated in our paper and acknowledged by reviewer **s7CG**, reinforcement learning results are highly stochastic, which compromises the reliability of the evaluation and may obscure the effectiveness of preference models. Hence, we do not perform RLHF experiments but employ rejection sampling, which produces more deterministic outcomes.
>
> According to your advice, we present additional rejection sampling results of our replicated AnthropicPM on the responses from Alpaca-7B and Vicuna-13B.
>
> | Model | Competitor | A win | Tie | B win |
> | :----: | :----: | :----: | :----: | :----: |
> | Alpaca-7B | **AutoPM** vs greedy | **45** | 8 | 27 |
> | Alpaca-7B | AnthropicPM vs greedy | 41 | 13 | 26 |
> | Vicuna-13B | **AutoPM** vs greedy | **46** | 13 | 21 |
> | Vicuna-13B | AnthropicPM vs greedy | 42 | 22 | 16 |
>
> The table above compares the responses selected by the preference models and the greedy decoded response. It demonstrates that our proposed AutoPM outperforms AnthropicPM in choosing more responses that are superior to the greedy decoded one. The table below directly contrasts the responses selected by AutoPM and AnthropicPM, where AutoPM prevails in more cases.
>
> | Model | Competitor | A win | Tie | B win |
> | :----: | :----: | :----: | :----: | :----: |
> | Alpaca-7B | **AutoPM** vs AnthropicPM | **23** | 37 | 20 |
> | Vicuna-13B | **AutoPM** vs AnthropicPM | **26** | 35 | 19 |
>
> Furthermore, AnthropicPM is trained with a large amount of manually labeled prompts and manually selected preference data, whereas AutoPM is trained with automatically generated data, which further attests to the superiority of AutoPM.
>
> ---
>
> ### **R4: Advice to add the human assessment of the paired model data generation.**
>
> | Paired Model Data Generation | Chosen > Rejected | Chosen ≈ Rejected | Chosen < Rejected |
> | :----: | :----: | :----: | :----: |
> | Ratio | 94% | 6% | 0% |
>
> Thanks for your suggestion. We have performed a human evaluation of the paired model data generation, which we initially omitted due to its triviality. We will include the experimental result in the future version of our paper and present it in the table above.
>
> For our experiment, we used text-davinci-003 to generate positive responses and OPT-IML-1.3B to generate negative responses. The OPT-IML-1.3B model is much smaller in size and has fewer training data than text-davinci-003, so there is no case where the OPT response is superior to text-davinci-003. The 6% of ties are due to text-davinci-003 producing incorrect answers. Overall, the preference data from the paired model data generation is valid for AutoPM training.
>
> ---
>
> ### **Q1: For IDDG, how to assemble multiple pairs of preference data?**
>
> The blue arrows in Fig. 1 and lines 285-288 (Section 3.2) of our paper describe how we assemble multiple pairs of preference data in IDDG: After executing one sequence of edit actions, we compose responses from two random intermediate steps. Specifically, after 1∼3 steps of edit action, we can obtain multiple responses with different quality, e.g., response A > response B > response C. Then we can generate three preference data pairs (A, B), (B, C), and (A, C). This assembly procedure is the same as in InstructGPT; however, they employ human annotators to rank the responses.
>
> ---
>
> ### **Q2: How are the examples in Appendix A.1 sampled?**
>
> We first randomly sample data from the dataset generated by In-breadth, In-depth, and paired models data generation. Then, we exclude any data that may contain inappropriate content (such as illegal, sexual, or hateful expressions). Therefore, the examples presented in this paper are not strictly uniformly sampled, but neither are they cherry-picked.
>
> ---
>
> ### **Q3: Which word-antonym pairs and insert/substitute/delete operations were applied for each of the generations in Appendix A.1?**
>
> Thanks for the suggestion. As the examples in Appendix A.1 are randomly sampled from the dataset, they are generated with different word-antonym pairs and edit actions. We will append these details for each example in the future version. Here is one example:
>
> > \# In-Breadth Data Generation with the phrase pair **(“correct” vs. “contains wrong information”)**.
> > **Prompt**: "Give two examples of a liquid."
> > **Chosen**: "Water and Oil."
> > **Rejected**: "Trees and Rocks."
>
>
> ---
>
> ### **Q4: In Figure 3 how can you get ties between 3 humans?**
>
> Given a pair of responses for each prompt, each human annotator is asked to select 'A win', 'Tie', or 'B win', therefore it is possible to produce ties. We will clarify this point in our paper.

---

### Official Review · Reviewer_s7CG · 2023-08-15

**Soundness:** 4

**Excitement:**

4: Strong: This paper deepens the understanding of some phenomenon or lowers the barriers to an existing research direction.

**Paper Topic And Main Contributions:**

Preference data (e.g., a human rating that LLM generation A is better than B on usefulness) can be used to make LLMs more aligned with human values. The paper proposes several prompt-based methods to automatically generate such preference data:

* In-breadth data generation: Use a prompt such as “Given the question, generate a [useful] answer A and a [useless] answer B” to generate pairs that align with one of the HHH criteria (helpful-honest-harmless).
* In-depth data generation: Use prompts to edit an LLM generation into a less preferred output, then pair two of the intermediate steps, with the earlier step being treated as positive.
* Paired models data generation: Use generations from two LLMs on the same prompt, with the generation from the stronger model being treated as positive.

The paper uses the generated data to train AutoPM, a score-based preference model. AutoPM is trained to score the positive output higher than the paired negative. The experiments compare AutoPM to (1) using a LLM to predict the preference, and (2) two previous preference models. It also demonstrates the use of AutoPM as a re-ranker at inference time.

**Reasons To Accept:**

1. The in-breadth and in-depth data generation methods explicitly target specific alignment criteria (helpfulness, honesty, and harmlessness). As such, the generated pairs are more likely to demonstrate the alignment criterion than random generations.
    * This could explain why AutoPM performs better than the baseline preference models of comparable sizes, despite using fewer preference samples.
    * For in-depth generation: while perturbation-based data synthesis is a common technique, the proposed method makes the edit actions correspond to the alignment criteria in a principled way.
    * The methods could be easily adapted to other criteria (politeness, creativeness, etc.).

2. The experiments are thorough.
    * (Section 5.3) The preference model is evaluated on 3 alignment criteria, using 3 different datasets. The experiments are compared with representative previous works.
    * (Section 5.4) The preference model is also shown to align well with human overall preference (not confined to the 3 criteria).
    * (Section 5.5) Finally, the preference model is shown to help improve LLMs by acting as a re-ranker. While it would have been great to see LLMs getting fine-tuned based on signals from the preference model, the results would have higher variance as the paper justifies.

**Reasons To Reject:**

1. The paper does not do much comparison between the 3 data generation methods.
    * This weakens the justification for each method. For instance, the paper claims that the in-depth method aims at generating a wider quality range than the in-breadth method (see the abstract and introduction), but this claim is not substantiated.
    * See also the last bullet in Weakness 2 below.

2. How much the amount of generated data affects the results is unclear.
    * The experiments use a fixed amount of generated preference data. This leaves some unanswered questions. For instance, is it possible to match the performance of stronger models like GPT-4 by synthesizing more data? If so, how much? Would a smaller number of prompts be enough?
    * If I understand correctly, the ablation studies also seem to use an unequal amount of data in each row. The increased numbers might be due to having more data. Considering the gains from in-breadth generation is bigger, maybe using in-breadth generation on the entire instruction set would have led to better results.

3. (minor) The in-breadth method relies on a small number of phrase pairs like [useful]-[useless] to construct the prompts. This might lead to artifacts in the data.
    * It would be interesting to see how the number of phrase pairs affect the results. For example, would adding more phrase pairs (and thus more prompts) lead to more data diversity, and thus a better preference model? Or do phrase pairs in the same category actually produce similar response pairs?

**Reproducibility:**

4: Could mostly reproduce the results, but there may be some variation because of sample variance or minor variations in their interpretation of the protocol or method.

**Reviewer Confidence:**

4: Quite sure. I tried to check the important points carefully. It's unlikely, though conceivable, that I missed something that should affect my ratings.

**Typos Grammar Style And Presentation Improvements:**

* Line 458: table 4 → Table 4
* Line 546: “prone to output the true statement” -- should this be “false statement”?

---

> ### Author Rebuttal · Authors · 2023-08-28
>
> We are grateful to reviewer **s7CG** for recognizing our work, as well as the valuable advice. We present our responses in the following.
>
> ### **R1: Advice to add more comparison between the 3 data generation methods.**
>
>
> | Data Composition | HHH Eval | TruthfulQA |
> | :---- | :----: | :----: |
> | In-Breadth *only* (42K) | 0.78 | 0.54 |
> | In-Breadth + Paired-Models *equally* (42K) | 0.80 | 0.54 |
> | In-Breadth + In-Depth + Paired-Models *equally* (42K) | **0.83** | **0.56** |
>
> Thank you for your valuable advice. We conducted an extra ablation study with a fixed number of training data (42K) to verify the contribution of each kind of data.
>
> (1) As shown in Table 7, the in-breadth data generation contributes the most to the performance of AutoPM. We provide the result using only in-breadth data generation to synthesize the overall 42K training data, which achieves a 2% improvement.
>
> (2) However, under the same amount of training data, replacing parts of in-breadth data with another two types of data can bring significant improvement. For the HHH evaluation, using in-breadth data generation only achieves an accuracy of 0.78, while using all three types of generated data with an equal size to comprise the overall 42K training data could achieve a higher accuracy of 0.83. This is because the paired models data generation can provide more natural preference data and the in-depth data generation can assemble preference data with various qualities. These two components further increase data diversity and are supplements to in-breadth data generation.
>
> From the ablation studies, we conclude that all three data components of AutoPM are crucial.
>
> ---
>
>
> ### **R2: Advice to analyze how much the amount of generated data affects the results.**
>
> | Data Amount| 10% | 20% | 30% | 50% | 100% | 200% |
> | :----: | :----: | :----: | :----: | :----: | :----: | :----: |
> | HHH Eval | 0.57 | 0.67 | 0.75 | 0.81 |0.83 | 0.83 |
> | TruthfulQA | 0.33 | 0.41 | 0.50 | 0.53 | 0.56 | 0.57 |
>
> Thanks for your valuable advice. Here we present the model performance trained by different amounts of generated data, where 100% corresponds to 42K data used in our paper. As the results show:
>
> (1) 50% to 100% of training data can achieve nearly the best performance, while a small portion (e.g., 30%) of training data exhibits a remarkable drop in performance.
>
> (2) For 200% of training data, we reuse prompts in the Alpaca dataset to generate a new set of preference data. We find that 200% training data brings only a slight boost since the generated preference data from the same prompt are similar to each other.
>
> ---
>
> ### **R3: How the number of phrase pairs affects the results.**
>
> \#Phrase Pairs | HHH Eval | TruthfulQA |
> :---- | :----: | :----: |
> 3 | 0.69 | 0.47 |
> 12 | **0.76** | **0.52** |
>
> Thanks for your advice. To collect the phrase pairs, we first harvest a wide variety of phrase pairs, then we filter the phrase pairs that are similar or hard to generate valid preference data, constituting the final 12 phrase pairs in our method.
>
> (1) In our preliminary experiments, we used only three phrase pairs ('helpful' vs 'unhelpful', 'honesty' vs 'dishonest', and 'harmless' vs 'harmful') to generate preference data. As shown in the table above, enriching the phrase pairs can indeed lead to a stronger performance.
>
> (2) Besides, we empirically observe that similar phrase pairs will produce similar preference data. Adding these data does not further strengthen the existing results. The 12 phrase pairs used are reserved after our careful construction. How to build a larger and more diverse phrase pair set is an open challenge, which can be an interesting future direction.

---

### Meta-Review · Area_Chair_govR · 2023-09-20

**Recommendation:** 5

**Metareview:**

This paper presents several strategies for automatically generating a dataset for training a human preference model. Reviewers found the methods well-motivated and effective. Reviewers also praised the thorough experiments covering multiple tasks, although the paper could benefit from further comparisons/analysis of the data generation strategies (reviewer s7CG) or discussion of alternative training methods (Dan7). Reviewer s7CG wanted to ensure the benefits of the strategies hold while controlling for dataset size and D9n7 suggested a few further experiments/numbers to report; the authors provided new results in their rebuttal which addressed the concerns. Although the intrinsic evaluation of the preference model itself is convincing, the only downstream evaluation on using it to improve LLM generation uses rejection sampling rather than RLHF. While this is a slight gap in the paper, I do agree with the authors’ point that RLHF is complex and potentially high-variance. A missing but relevant reference is Constitutional AI (https://arxiv.org/abs/2212.08073), which I encourage the authors to discuss in the paper. Overall I agree with reviewers that the method is solid and well-justified empirically.

---

### Decision · Program_Chairs · 2023-10-07

**Decision:**

Accept-Main

**Comment:**

This paper presents several strategies for automatically generating a dataset for training a human preference model. Reviewers found the methods well-motivated and effective. Reviewers also praised the thorough experiments covering multiple tasks, although the paper could benefit from further comparisons/analysis of the data generation strategies (reviewer s7CG) or discussion of alternative training methods (Dan7). Reviewer s7CG wanted to ensure the benefits of the strategies hold while controlling for dataset size and D9n7 suggested a few further experiments/numbers to report; the authors provided new results in their rebuttal which addressed the concerns. Although the intrinsic evaluation of the preference model itself is convincing, the only downstream evaluation on using it to improve LLM generation uses rejection sampling rather than RLHF. While this is a slight gap in the paper, I do agree with the authors’ point that RLHF is complex and potentially high-variance. A missing but relevant reference is Constitutional AI (https://arxiv.org/abs/2212.08073), which I encourage the authors to discuss in the paper. Overall I agree with reviewers that the method is solid and well-justified empirically.